# The Role of Blood Pressure Load in Ambulatory Blood Pressure Monitoring in Adults: A Literature Review of Current Evidence

**DOI:** 10.3390/diagnostics13152485

**Published:** 2023-07-26

**Authors:** Ophir Eyal, Iddo Z. Ben-Dov

**Affiliations:** Department of Nephrology and Hypertension, Hadassah Medical Center and Faculty of Medicine, Hebrew University of Jerusalem, Jerusalem 9112001, Israel; iddo@hadassah.org.il

**Keywords:** hypertension, ABPM, blood pressure load, target organ damage, LVH

## Abstract

Background: The blood pressure load (BPL) is commonly defined as the percentage of readings in a 24-h ambulatory blood pressure monitoring (ABPM) study above a certain threshold, usually the upper normal limit. While it has been studied since the 1990s, the benefits of using this index have not been clearly demonstrated in adults. We present the first review on the associations of BPL with target organ damage (TOD) and clinical outcomes in adults, the major determinants for its role and utility in blood pressure measurement. We emphasize studies which evaluated whether BPL has added benefit to the average blood pressure indices on ABPM in predicting adverse outcomes. Methods: PubMed search for all English language papers mentioning ABPM and BPL. Results: While multiple studies assessed this question, the cumulative sample size is small. Whereas the associations of BPL with various TODs are evident, the available literature fails to demonstrate a clear and consistent added value for the BPL over the average blood pressure indices. Conclusions: There is a need for prospective studies evaluating the role of BPL in blood pressure measurement. The current literature does not provide sound support for the use of BPL in clinical decisions.

Twenty-four-hour ambulatory blood pressure monitoring (ABPM) has become an increasingly applied tool in both the diagnosis of hypertension and in the assessment of its treatment response. While the main indices derived from ABPM are the systolic and diastolic averages of the time periods of interest (24 h, awake time and asleep time), these do not reflect the variability of blood pressure (BP). Of the different methods developed to estimate variability in an ABPM study, one of the most popular indicators has been the blood pressure load (BPL). Though a few studies used this term in different way, it is most commonly defined as the percentage of BP readings above a certain value, usually the upper limit of normal ambulatory BP, a concept introduced in the late 1980s [1,2]. Figure 1 shows the stable publication rate concerning BPL over the past three decades. However, the clinical importance of the BPL in adults has not been clearly elucidated, and it has generally not been integrated into adult hypertension guidelines such as those of the ACC/AHA [3], Hypertension Canada [4] or NICE [5]. The European Society of Hypertension (ESH) guidelines on ABPM performance stipulate BPL should be included in the research report but not the clinical report [6], and BPL is not mentioned in the ESH hypertension guidelines [7,8]. Past Australian guidelines mentioned BPL [9] only to omit it in newer revisions [10]. Despite this, BPL has been included in the standard reports produced by commonly used ABPM software. Given this significant level of exposure, it is quite possible that BPL has been used in clinical decision making, though we found no studies exploring this.

As the diagnosis and treatment goals of hypertension are sought to be defined as threshold values for the prevention of morbidity and mortality, earlier signs of target organ damage (TOD) are frequently used as proxies. We hereby report a literature review on the associations of BPL with TOD and cardiovascular (CV) outcomes in adults. Our review was based on a search of PubMed for all articles containing the combination of the MeSH term “Blood Pressure Monitoring, Ambulatory” and “blood pressure load” or “BP load”, last performed on April 13th, 2023. We also scanned the references from articles found on the PubMed database. We included all the articles for which we could find the full text in English examining the associations of BPL as defined above with clinical outcomes or organ damage in adults.

Of note, the progression in the ABPM technique from fixed awake/sleep-hours definitions to flexible settings, adjusted according to the patient activity diary, introduces difficulty in integrating results from the older and newer studies. More importantly, as BPL is defined by values above the upper limit of the normal ambulatory BP, the lowering of these thresholds in more recent guidelines creates room for even more inconsistencies in the knowledge base.

In almost all of the studies described below, both BPL and the various TODs were evaluated together at a single point measurement. A few exceptions employed a one-time ABPM with a prospective follow-up of TODs, and these are also detailed below.

## 1. Left Ventricular Hypertrophy

Left ventricular hypertrophy (LVH) is a common TOD of hypertension, well associated with cardiovascular morbidity and mortality. LVH, typically assessed using the left ventricular mass index (LVMI) measured by echocardiography, is perhaps the most evaluated TOD in the context of BPL. In a pioneering study, White et al. [11] examined 30 patients (mean age 47 ± 12 years, 57% men) with mild–moderate essential hypertension and without significant comorbidities. BPL was defined by the threshold values of 140/90 mmHg and 120/80 mmHg for awake and asleep BP, respectively. These were chosen based on the findings of a previous study, in which this cutoff produced BPL values of less than 10% (awake period) and 5% (asleep period) in the normotensive participants (BP < 135/80 per office measurement). The authors report an average BPL of ~45% for the various BPL indices (awake/asleep, systolic/diastolic). LVMI showed a significant positive correlation with both 24-h systolic BPL (SBPL, r = 0.68) and 24-h average systolic BP (ASBP, r = 0.54). The significance of the difference in the correlation coefficients was not reported. In this small study, an SBPL lower than 40% and diastolic BPL (DBPL) lower than 50% predicted less than 10% risk for LVH. The left ventricular filling rate, assessed by ventriculography, was significantly inversely correlated with both 24-h average SBP and diastolic BP (ADBP), as well as with SBPL and DBPL. SBPL and DBPL lower than ~40% predicted less than 10% risk of slow filling. Multivariate analyses on the correlation of BPL and average blood pressure (ABP) with LVH were not performed.

Over the subsequent three decades, other studies have explored the possible association between BPL and LVH. Several reports [12,13,14,15] included less than 100 participants, and one study [16] included 335, mostly with an average age of the fifth decade of life and without significant comorbidities or antihypertensive treatment. The thresholds applied for BPL were 140/90 mmHg and 120/80 mmHg for awake and asleep BP, respectively, and awake/asleep limits were fixed when reported. Generally, these studies found associations between LVMI and BPL, with variations between studies in the best correlated index (for example DBPL [13] or asleep BPL [15]). These studies also reported correlations between LVMI and ABP parameters from ABPM, which were of similar magnitude to the correlations with BPL. Polónia et al. [12] reported stepwise multiple regression in which only age, awake ASBP and BPL independently correlated with LVMI. Grossman et al. [14] found by stepwise regression that DBPL was the major determinant of LVMI with r = 0.5. In the study by Tsioufis et al. [16], multivariate regression analysis found 24-h SBPL to be correlated with LVMI; however, ABP indices from ABPM were not included in the model.

Several studies have aimed to better assess the contribution of BPL over the indices of ABP in ABPM. Nobre et al. [17] studied 1143 patients (mean age 52 years, 54% men) at a single center in Brazil, more than half of whom were treated with anti-hypertensives, with ABPM performed to assess BP control. BPL threshold limits were 140/90 and 120/80 for awake and asleep time, respectively, and awake/asleep times were fixed. For the entire cohort, 24-h SBPL and DBPL were very strongly correlated (r = 0.90) with 24-h ASBP and ADBP, respectively. When limiting the comparison to participants with BPL of 90–100%, the correlation weakened to r values ~0.5, which was expected as the much smaller range of BPL limited the discrimination ability. Among the 329 patients for whom echocardiography was available, 24-h ASBP and ADBP were significantly correlated with LVMI with r values of 0.56 and 0.35, respectively, and SBPL and DBPL were similarly significantly correlated with LVMI with r values of 0.49 and 0.33, respectively. When limiting the comparison to participants with BPL of 90–100%, no significant correlation was found. To summarize, 24-h SBPL and DBPL were highly correlated with their respective 24-h averages, and the significant correlations observed with LVMI were of similar magnitudes. In considering the utility of elevated BPL to identify TOD risk in people with normal ABP, when examining the 97 participants with 24-h ASBP < 135 mmHg but SBPL above 50%, no correlation was found with LVMI.

Mulè et al. [18] aimed to identify whether, among patients with similar ABP, higher BPL portends more TODs. To approach this, they investigated 130 participants in Italy (mean age 48 ± 1 years, 70% men) with primary hypertension and without antihypertensive treatment either at baseline or temporarily withheld for the study. BPL thresholds were 140/80 and 120/80 for awake and asleep times; awake/asleep hours were adjusted individually. As expected, regression equations found very strong correlation between corresponding systolic and diastolic 24-h ABP and BPL. Due to collinearity, multiple regression analysis was not feasible, and patients were categorized into high or low BPL groups based on whether their BPL exceeded that predicted by their ABP. Comparison of SBPL groups showed no difference between the two groups in LVMI, but other cardiac parameters were worse in the high-SBPL group. DBPL groups did not differ in any of the cardiac parameters.

Tatasciore et al. [19] assessed BPL as part of their study which included 180 patients (mean age 53 ± 8 years, 60% men) with a recent (less than 6 months) diagnosis of hypertension and without significant comorbidities. BPL thresholds were 135/85 for awake time; awake/asleep hours were adjusted individually. SBPL and DBPL were significantly associated with LVMI in univariate models, but this association was lost in a multivariate model including ABP and several clinical parameters.

Liu et al. [20] examined various TODs in 869 patients (mean age 51 ± 11 years, 50% men) with suspected hypertension from a single center in China, of whom 460 had echocardiography results. ABP was time weighted according to the interval between readings. BPL thresholds were 135/85 and 120/70 for awake and asleep times; awake/asleep hours were fixed. Analyses were performed using BPL tertiles to adjust for a skewed distribution, and also using BPL as a continuous variable, excluding those with BPL of 0%. In order to evaluate the relative contributions of BPL and ABP, multivariate regression models for the different TODs were built in pairs—one using the BPL and the second using 24-h ABP. Stepwise addition of the other variable (i.e., either ABP or BPL) allowed for the assessment of improvement in the model’s goodness of fit. Of note, compared to the first tertile of BPL, participants in the third tertile were slightly younger but had multiple more cardiovascular risk factors including higher BMI, male sex, smoking status and alcohol use and, unsurprisingly, had a higher 24-h ASBP (140.4 vs. 114.5). When considered in tertiles, higher BPL was significantly associated with higher LVMI even after adjustment for traditional risk factors. However, when ABP was included in the model, the association was lost. With BPL as a continuous parameter, the addition of SBPL did not improve a multivariate model that included ABP. Inversely, the addition of ABP significantly improved the predictive power of a model that included BPL. Separate analyses for patients with ambulatory hypertension vs. those with high-normal BP, or separating BPL for awake time and asleep time, yielded the same results. The authors summarize that LVH (and other TODs, see below) was correlated with high BPL, but this was not independent of the ABP, and this was consistent in all analyses including patients with high-normal BP. The authors acknowledge that partition to tertiles could have lacked sensitivity and that the population studied was relatively low risk.

## 2. Special Populations

Wang et al. [21] explored BPL in 32 peritoneal dialysis patients in Taiwan using the 140/90 mmHg and 120/80 threshold values and fixed awake/asleep limits. They report correlations of LVMI with both asleep ABP values and asleep SBPL and DBPL, all with r ≈ 0.45. Categorizing BPL as above or below 30% resulted in a slightly weaker correlation. Multivariate logistic regression incorporating multiple parameters (patient-related, dialysis-related, blood pressure indices) found only SBPL above 30% to be independently associated with LVH.

Toprak et al. [22] examined the association of LVMI with BPL in 35 kidney transplant recipients in their mid-thirties in Turkey. BPL was defined by 135/85 and 125/75 for awake and asleep values, and awake/asleep hours were set individually. In a multivariable regression analysis, asleep SBPL and hemoglobin were the only independent variables associated with LVMI. However, ABP values were not included as parameters in this regression.

Wang et al. [23] examined the association of LVMI with BPL in chronic kidney disease (CKD) patients in a single center in China. A total of 1219 patients (mean age 44 ± 17 years, 59% men) participated, of whom 432 (35.4%) had non-diabetic CKD without hypertension, 565 (46.3%) had non-diabetic CKD with hypertension and 222 (18.2%) had diabetes, with increases between the groups in their order of mentioning in a wide range of clinical and biochemical cardiovascular risk factors. BPL was defined by 135/85 and 120/70 for awake and asleep times, with awake/asleep hours set individually. In all three groups, while univariate analyses showed correlations between some BPL indices and LVMI, multivariable-adjusted models that included 24-h ABP found the addition of the relevant BPL index was negligible (R^2^ = 0.034, R^2^ = 0.008 and non-significant for the three groups, respectively).

Jaques et al. [24] examined the association of LVMI with BPL in 69 patients in their sixties and seventies with advanced chronic kidney disease (KDIGO stage 3b-5) who were not receiving dialysis. BPL was defined by 135/85 and 125/75 for awake and asleep times, with awake/asleep hours set individually. In univariate analysis, LVMI was associated with systolic non-dipping status and mean BP non-dipping status, as well as with 24-h BPL and asleep time SBPL. However, these associations with BPL were lost in backward stepwise multivariate analysis. For patients with hypertension per ABPM (defined as 24-h ABP ≥ 130/80 mmHg), increased LVMI was associated with only three ABPM parameters—24-h DBPL, awake SBPL and awake DBPL. The authors also examined associations with LVH (defined as an absolute higher myocardial mass), and in a multivariate analysis, 24-h and asleep DBPL were negatively associated with LVH. The authors concluded that BPL did not show consistent association with LVH beyond ABP. A total of 56/72 (78%) of the patients had repeated echocardiography after 1 year; an association of BPL with increased LVMI was not found at this time point as well.

A summary of the above-mentioned studies exploring the associations of LVH with BPL is presented in Table 1.

## 3. Other Target Organ Damages

Blanco et al. [25] explored the association of BP with two markers of carotid atherosclerosis—the common carotid intima-media thickness (CCIMT) and common carotid diameter (CCD). They report 292 participants (mean age 73 ± 6 years, 45% men), most of whom were hypertensive but less than half of these were treated and only 17.8% well controlled per office BP. BPL threshold limits were 140/90 and 120/80 for awake and asleep times; awake/asleep hours were set individually. In a multivariate model, the strongest association for CCIMT was with the sleep pulse pressure (r ≈ 0.32) and for CCD with the 24-h SBPL (r ≈ 0.45).

De la Colina et al. [26] assessed the association between the BPL and cognitive function, hypothesizing that surges in BP are deleterious. They examined 49 normotensive participants and 28 with controlled hypertension, aged 60–75 years and without significant comorbidities, employing ABPM and a battery of neuropsychological tests. BPL thresholds were 130/80 for 24 h or 135/85 for awake time. A total of 32.6% in the normotensive group had SBPL and DBPL of 0%, in contrast to only 10.7% in the controlled hypertension group. After adjusting for several covariates, awake SBPL was significantly correlated with several cognitive parameters in both groups, while awake DBPL had such associations only in the controlled HTN group. No such association was found with asleep BPL. The 24-h SBPL also showed stronger associations in the controlled hypertension group than the normotensive group, while no association was found for 24-h DBPL. The authors conclude that episodes of high BP possibly damage small cerebral blood vessels, leading to decreases in various cognitive functions. However, such damage could have occurred before hypertension was controlled, unrelated to the difference in BPL. Even more importantly, while ASBP and ADBP were within the target range in both groups, they were higher in the controlled hypertensives (123/72 vs. 116/70), and a model adjusting for that difference was not employed.

In the study by Mulè et al. [18] described above, the group with higher SBPL experienced significantly more retinopathy, but no difference was observed in creatinine or albuminuria. No such association was found for DBPL.

In the study by Liu et al. [20] described above, TOD parameters included urinary albumin/creatinine ratio, carotid–femoral pulse wave velocity and brachial–ankle pulse wave velocity. The results for these measures were on a par with those for LVH, i.e., higher BPL values were associated with more TODs, but in multivariate analyses which included ABP parameters, the addition of BPL was non-contributory.

The study by Wang et al. [23] on CKD patients described above also evaluated CCIMT, kidney function per the MDRD equation and proteinuria measured via 24-h urine collection. Similarly to the findings for LVMI (see above), even when a significant correlation was found with a BPL index in a univariate model, the contribution of this index to a multivariable model which included ABP was negligible if at all significant.

## 4. Cardiovascular Outcomes

Andrade et al. [27] explored the prognostic importance of BPL in 126 patients (63% women) aged over 80 years with hypertension treated to <140/90 in a single center in Brazil. BPL was defined by 135/85 and 120/80 for awake and asleep times, and awake/asleep hours were fixed. Mean follow-up was 23 ± 5.6 months, during which 12 cardiovascular events were noted (9.5% of the sample); those patients had a significantly higher prevalence of previous strokes (33.3% vs. 7.0%, *p* = 0.02). Additionally, the patients with CV events had significantly higher awake ASBP (138 vs. 126) and SBPL (44.2% vs. 20.6%), asleep ASBP (133 vs. 121) and SBPL (71.6% vs. 47.2%). However, in a multivariable analysis, only two variables associated independently with CV events—a history of stroke and awake SBPL ≥ 24.5%. An SBPL lower than this value had a negative predictive value of 98% in this cohort.

Li et al. [28] aimed to assess the added value of BPL over ABP in predicting the risk of cardiovascular events using the International Database on Ambulatory BP in relation to Cardiovascular Outcomes (IDACO), which includes several prospective population cohorts. The analysis included 8711 patients from 10 cohorts of whom 62% were European, 21.5% Asians, 16.5% South Americans and 47% were women. BPL was defined by 135/85 and 120/70 for awake and asleep times, with fixed awake/asleep hours definitions. BPL and ABP correlated very strongly for both SBP and DBP (r ≥ 0.91 and r ≥ 0.88, respectively). In a multivariate Cox model that included typical risk factors, ASBP and ADBP predicted overall mortality and cardiovascular mortality and the combined fatal and non-fatal cardiovascular events. The addition of SBPL or DBPL to these models did not improve them in a significant and meaningful way. Consideration of BPL as a categorical parameter yielded the same results.

In another study, Li et al. [29] focused on the prognostic value of asleep SBPL and DBPL in stable CKD patients. The study included 588 participants (mean age 43 ± 17 years, 57% men), of whom 400 had chronic glomerulonephritis and only 63 had diabetic nephropathy. Notably, less than 25% had stage 4 or 5 CKD. BPL was defined by 135/85 and 120/70 for awake and asleep times, and awake/asleep were defined individually. The patients were divided into tertiles of SBPL with a parallel analysis for DBPL. Importantly, patients in the third tertile had significantly more risk factors and also higher BP (clinic, 24 h, awake), despite more antihypertensive medications. The primary outcome was all-cause or cardiovascular mortality, and secondary outcomes included renal events and several cardiovascular events. For both SBPL and DBPL, patients in the higher tertiles experienced significantly more primary and secondary outcomes. Multivariate regression also found the third tertile to be a risk factor compared to the first tertile; however, ABP was not included in the adjusting parameters.

Tikhonoff et al. [30] examined BP as a predictor of incident atrial fibrillation (AF). Their cohort included 3956 patients aged ≥ 18 years (48% men) in Europe, 65% Flemish, with median follow-up of 14.0 years. A total of 62.0% of the participants were normotensive, 15.1% had treated hypertension, 8.1% untreated sustained hypertension, 9.2% masked hypertension and 5.5% white-coat hypertension. BPL was determined only by SBP above 135 mmHg during awake time; awake/asleep hours were fixed. In 2776 participants with complete 24-h ABPM, ASBP for the 24-h, awake or asleep periods as well as 24-h ADBP all predicted new AF. When including the rest of the cohort, which only had awake time ABPM data, the results were similar. SBPL was considered in quartiles as 20% had BPL of 0%; patients in increasingly higher quartiles had more risk factors including higher age and male sex. The risk of incident AF increased significantly with the rise in quartiles. The authors did not address whether SBPL improved upon the predictions of mean SBP.

## 5. Discussion

The concept of BPL was introduced 35 years ago, and it has since garnered considerable attention in research. Though it has not been adopted in treatment guidelines, the inclusion of BPL in standard ABPM reports quite possibly leads to it influencing clinical decisions. The motivation in exploring this parameter was based on the realization that ABP cannot reflect BP variability, considered a sign and perhaps a harbinger of cardiovascular damage [31], and on the assumption that spikes in BP could be deleterious. As BPL has a skewed distribution [20], it is expected to incompletely correlate to ABP and thus to provide additional information on blood pressure control. The employment of BPL as a routine ABPM index is called for if it improves the utility of the ABPM in guiding treatment decisions, either as an additional criterion in the diagnosis of hypertension or as a constituent of treatment targets. The studies described above used various non-interventional designs and populations to assess the association between BPL and signs of TOD, mainly LVH, or occurrence of cardiovascular outcomes. As mentioned above, the studies differed in their approach to classifying readings as awake or sleep time; obviously the older technique of fixed awake and sleep time windows could affect the results. Additionally, as changes were made to the blood pressure thresholds above which a reading was included in the BPL, contemporary reanalysis of the older trials could produce different conclusions.

Beyond this lack of uniformity, the studies reviewed share several other limitations. First, apart from 2 studies with ~8700 [28] and ~4000 [30] participants, all other studies combined included ~5500 participants. This is a modest sample size, especially compared to modern investigations in the field of hypertension. Most of these studies excluded patients with significant comorbidities, also limiting the ability to generalize the results to a large group of high risk patients in whom refined risk assessment is particularly important. Moreover, ABPM was not repeated, and evidence for the reproducibility of BPL are scant and short term [32]. Of note, the assessment of multiple BPL parameters (both systolic and diastolic for three time periods) leads to multiple comparison testing and thus increases the risk of a type 1 error.

Moreover, almost all the studies were retrospective in nature, as they examined TODs that were prevalent at the time of ABPM but had developed over a previous period. Except for one small study [24], the markers of TODs were measured at a single time point, precluding assessment of the effect of BPL over time. Even more importantly, a major disadvantage of the current knowledge base is the universal use of non-interventional designs, i.e., we found no studies that assessed whether treatment according to BPL goals would achieve superior clinical outcomes to treatment by ABP. To achieve sufficient statistical power, such a trial would require a substantial number of participants and prolonged follow-up.

Despite these limitations, it can be summarized that significant and meaningful associations exist between BPL and the studied outcomes, though the results varied in terms of the specific BPL index, whether systolic or diastolic and for which time period. Disappointingly, however, numerous studies failed to show a consistent advantage in using the BPL alongside the ABP indices. Of the 11 studies described above which utilized multivariable models that incorporated ABP parameters, only 5 found BPL parameters to have a contributing effect. Moreover, these were some of the smaller studies, and the significant BPL parameters differed among them. Importantly, the studies which more rigorously tackled this particular question did not demonstrate an additional benefit from the use of BPL over the ABP. This holds true for newly diagnosed patients, those with established hypertension and patients who were found to be normotensive by ABPM. Of note, in the pediatric population, the 2022 update of the AHA for pediatric ABPM [33] eliminated BPL from its criteria as it did not improve prediction of LVH compared to ABP [34].

An inherent flaw in BPL is that it sums all readings that exceed the defined values, but without weighting the degree of excursion above said values. Some have attempted to solve this by considering the area under the blood pressure curve [17,28,35], but this method is beyond the scope of this review.

In conclusion, while the BPL has been studied for more than 30 years, the available evidence fails to show that it offers an advantage over the use of ABP alone in adults undergoing 24-h ABPM. However, the current body of evidence has significant limitations. We call for dedicated studies in this field with appropriate statistical power and methods to determine whether an independent role exists for the blood pressure load index. Of particular interest is whether a high BPL alone should be taken as an indication to initiate or intensify treatment, i.e., in patients who are normotensive or with controlled hypertensives per ABP.

## Figures and Tables

**Figure 1 diagnostics-13-02485-f001:**
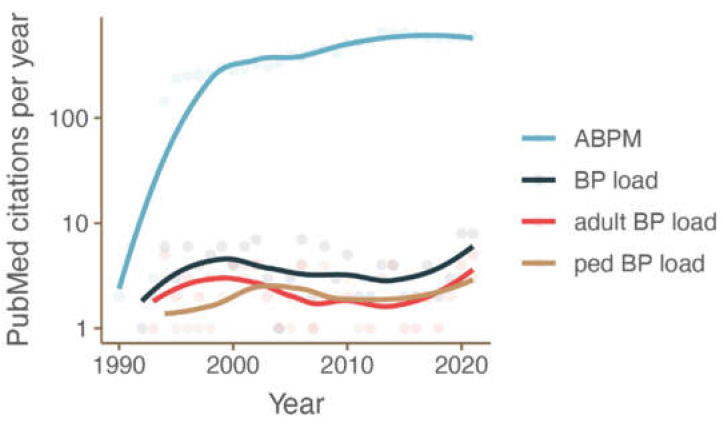
Number of PubMed citations per year for ABPM (blue) and for articles mentioning ABPM and BPL, with (red and brown) or without (black) age limits (age cutoff 18 years).

**Table 1 diagnostics-13-02485-t001:** Studies exploring association of blood pressure load with left ventricular hypertrophy.

Study	*n* of Participants	Mean Age [Years]	Men (%)	Population Studied	Threshold for Awake Period Load [mmHg]	Threshold for Asleep Period Load [mmHg]	Awake/Asleep Hours	Univariate Analyses	Multivariate Analyses
White 1989 [11]	30	47 ± 12	57	Mild–moderate untreated essential HTN without significant comorbidities.	140/90	120/80	Fixed	LVMI correlated with 24-h SBPL and 24-h ASBP.	N.A.
Polónia 1992 [12]	160	50	100	Highly sedentary, untreated, without significant comorbidities. Three groups with different SBP at rest and peak exercise.	140/90	120/80	Fixed	LVMI correlated with awake ASBP and SBPL.	LVMI correlated with awake ASBP and SBPL.
Bauwens 1992 [13]	35	46	63	Newly diagnosed untreated HTN without significant comorbidities.	140/90	140/90	N.A.	LVMI correlated with 24-h ASBP, ADBP, SBPL and DBPL	N.A.
Grossman 1994 [14]	60	40 ± 2	78	Mild–moderate untreated HTN without significant comorbidities.	140/90	140/90	Fixed	LVMI correlated with ASBP, ADBP, SBPL and DBPL.	DBPL main predictor of LVMI.
Musialik 1998 [15]	30	53	N.A.	Known HTN patients.	N.A.	N.A.	N.A.	Asleep MAP and BPL both correlated with LVMI in older patients.	Not performed.
Tsioufis 1999 [16]	335	52 ± 12	57	Known untreated essential HTN without significant comorbidities.	140/90	120/80	Fixed	Concentric LVH associated with higher 24-h ASBP, ADBP, MAP, SBPL and DBPL.	24-h SBPL associated with LVMI, but unclear which other indices were included in the model.
Nobre 2005 [17]	329	54	52	All patients referred for ABPM: half for diagnosis, half with known treated HTN.	140/90	120/80	Fixed	LVMI correlated with 24-h ASBP, ADBP, SBPL and DBPL with similar magnitudes. For patients with normal 24-h ASBP and SBPL > 50%, no correlation with LVMI.	N.A.
Mulè 2001 [18]	130	48 ± 1	70	Primary HTN without treatment.	140/80	120/80	Individual	24-h ABP and BPL highly correlated.	No difference in LVMI between groups with BPL higher or lower than expected per ABP.
Tatasciore 2007 [19]	180	53 ± 8	60	Newly diagnosed HTN without significant comorbidities.	135/85	NA	Individual	SBPL and DBPL associated with LVMI	SBPL and DBPL not correlated with LVMI in model including ABP and clinical parameters.
Liu 2013 [20]	460	51 ± 10	50	Suspected HTN.	135/85	120/70	Fixed	Higher BPL tertile associated with higher LVMI.	ABP improved the prediction of LVMI in a model including BPL, while BPL did not improve a model including ABP.
Wang 2001 [21]	32	NA	NA	Peritoneal dialysis patients.	140/90	120/80	Fixed	LVMI correlated with asleep ASBP, ADBP, SBPL and DBPL	Only asleep SBPL > 30% associated with LVH.
Toprak 2003 [22]	35	35	71	Kidney transplant recipients.	135/85	125/75	Individual	NA	Asleep SBPL associated with LVMI, but analysis did not include ABP.
Wang 2015 [23]	12,198	44 ± 17	59	Non-diabetic CKD with or without HTN, diabetics.	135/85	120/70	Individual	Several BPL indices correlated with LVMI.	BPL non-contributory to models including 24-h ABP.
Jaques 2018 [24]	69	69	65	Advanced non-dialysis dependent CKD.	135/85	125/75	Individual	LVMI correlated with 24-h BPL and asleep SBPL.	Association absent. Considering only hypertensive patients. increased LVMI associated with 24-h DBPL and awake SBPL and DBPL.

24-h—twenty-four-hour; ABP—average blood pressure; ABPM—ambulatory blood pressure monitoring; ADBP—average diastolic blood pressure; ASBP—average systolic blood pressure; BPL – blood pressure load; CKD—chronic kidney disease; DBPL—diastolic blood pressure load; HTN—hypertension; LVH—left ventricular hypertrophy; LVMI—left ventricular mass index; MAP—mean arterial pressure; N.A.—not available; SBP—systolic blood pressure; SBPL—systolic blood pressure load.

## Data Availability

Not applicable.

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
