# Peer review of "The Role of Blood Pressure Load in Ambulatory Blood Pressure Monitoring in Adults: A Literature Review of Current Evidence"

_diagnostics, 2023, doi:10.3390/diagnostics13152485_

Round 1

Reviewer 1 Report

I read with great interest the narrative review entitled “The Role of Blood Pressure Load in Ambulatory Blood Pressure Monitoring in Adults: A Review of Current Evidence”. The effort of the authors and the meticulous and extensive bibliographical review are appreciated, however, this narrative review does not contribute with innovative information to what is already known.

Author Response

We wish to thank the reviewer for the interest and time spent reading our work on the blood pressure load. Our paper indeed does not include new data, as is the nature of reviews. However, our paper is novel in that to the best of our knowledge, it is the first review on this subject. We believe this subject has long deserved an in-depth summary of the available studies, so that the hypertension community can better decide how to apply this index.

Reviewer 2 Report

This is an interesting narrative Review article focusing on the clinical significance of the so called BP Load whose prognostic value is still debated. The article is well written and provide useful information.

Taking into account the below comments and suggestions could improve the scientific impact of this study.

Throughout the text and tables, the authors should distinguish between mean BP (DBP+1/3 PP) and average ambulatory BP (avoid the use of the term “mean” when relating to average BP).

Other TOD data should follow LVH data in a section devoted to TOD, before the CV event section.

A summary table for CV events would be welcome.

Discussion and Conclusions. The value of the studies that did not control for average ambulatory BP should be downplayed as they have little clinical importance. The authors should focus on the studies that controlled for average ambulatory BP. It would be interesting to know the proportion of the studies controlling for average ambulatory BP that found a BPL prognostic capacity on top of average BP (probably a minority). The conclusion should be based on this finding.

The sentence “To achieve sufficient statistical power, such a trial would require a substantial number of participants and prolonged follow-up, but hopefully this challenge would be undertaken” should be rephrased.

Were BP variability indices included in any multiple regression? If not, this is another weakness of the studies on BPL.

The legend to figure 1 is unclear. Please provide better explanation for the single curves.

Ref 7. Update including new 2023 ESH Guidelines.

Update ref 31 by Parati et al. to the 2023 article.

Reviewer 3 Report

Dear Editor,

I carefully read the manuscript by Eyal et al.

Even though the manuscript seems to be very interesting, review's design needs to be improved.

My comments and suggestions for the authors are the following:

 - The authors should revise their article in the light of the new ESH guidelines recently released during the Annual Meeting of the European Hypertension Society.

 - Did the authors perform a systematic review? In this case, review's design should be clearly declared, according to the PRISMA guidelines. Moreover, the latest version of PRISMA guidelines should be included among the References.

 - In the case of a systematic design, a Table detailing the risk of publication bias for each study should be included in the manuscript, according to PRISMA guidelines.

Some typos need to be corrected.

Reviewer 4 Report

I read with interest the paper by Eyal and Ben-Dov titled ”The Role of Blood Pressure Load in Ambulatory Blood Pressure Monitoring in Adults: A Review of Current Evidence”.

I thank the authors and the editor for the opportunity to review this manuscript.

However, I have some reservations as detailed below.

Major points

In their literature review, the authors evaluated the literature on the association of blood pressure load (BPL) and target organ damage (TOD) and clinical outcomes.

The BPL is the percentage of values of an ambulatory blood pressure above a certain limit, e.g. the upper normal limit.

The sample size of most studies was small. The authors found that BPL was associated with TOD / adverse outcomes, primarily LVH. However, but BPL did not improve the prediction of adverse events over standard mean BP measurements in most of the studies and particularly there was no effect of BPL in multivariate analysis in the larger studies. A lot of the studies did not use multivariate analysis whereby it’s unclear whether BPL offers any utility over mean BP.

The authors conclude that further studies are needed. However several of the studies are quite large at 800-1000+ patients and generally do not find a significant effect of BPL in addition to mean BP.

The authors should rewrite the conclusion to just state that the evidence does not support use of BPL or that the evidence suggest that BPL adds little or no clinical value for the prediction of TOD. The readers can then assess by themselves whether more studies are needed on the subject.  

The manuscript is a literature review, not a systematic review which should be stated clearly. Thus, PRISMA guideline for systematic reviews was not used and there is no evaluation the level of evidence. The authors searched PubMed and the references of articles. However, the pubmed search could have been broadened, e.g. the MeSH term “Blood Pressure Monitoring, Ambulatory” seems not to include for example “24 hour blood pressure” which when added to the mesh term increases the number of hits by about 10%.

The design of the majority of the included studies are not described (single measurement for the association of BLP and TOD vs. follow-up over x years; Prospective vs. retrospective; Observational vs. interventional).

There are minor grammatical issues.I have commented on some of them below.

See specific points below.

Title

Please add “A literature review of current evidence”.

Abstract

”upper limit of norm” – should be ”upper normal limit”

Please specify that you evaluated 24 hours ambulatory blood pressure rather than the broader term Ambulatory Blood Pressure Monitoring.

it has been in general” – BPL may have been proposed in the 1990s but is not used in clinical practice (although it may be reported in the automatic evaluation).

“exploring whether BPL has added benefit to the mean blood pressure indices on ABPM.” - the sentence is long and the wording somewhat unclear. Consider to divide the sentence in two and, try e.g. “evaluated whether addition of BPL to mean blood pressure indices in ABPM improves the prediction of adverse outcomes.” or similar wording.

If there is room, consider to add the major types of TOD that were evaluated.

It may be noted that mean BP and BPL are strongly correlated.

Introduction

“popularized to the degree it is a part of the standard report produced by commonly used ABPM software” – Are there any evidence to suggest that BPL is used in clinical practice? BPL is not part of any clinical guidelines. Please rephrase and avoid the word popularized.

The use of BPL seems to be primarily in the scientific domain.

Did the authors have any exclusion criteria, patients with x conditions?

Left Ventricular Hypertrophy

Please clarify whether the studies evaluated e.g. only patients at a single time point and evaluated the correlation of BPL and LVH or whether these patients were free from LVH at baseline then followed the patients and evaluated the association between BPL and development of LVH and whether patients were treated.

This can be stated at least for the larger studies or it can be stated in a single sentence that e.g. some studies evaluated the association in x manner and theses and these studies evaluated the association in z manner.  

In table 1, please add a column for the type of study i.e.: “point measurement” vs. “follow-up”, prospective vs. retrospective, the number of months of follow up, whether hypertension was treated. 

“less than 10% and 5%” – unclear, why are there two different values for one group? This sentence on a previous study of the one in question seems redundant.

Regarding the study of White et al. (11) the authors have not clearly stated whether addition of BPL to mean BP changes prognostication. Was there an analysis of the effect of BPL corrected for mean BP? Or were the correlations significantly different?

“To summarize, 24-hours SBPL and DBPL” – superfluous.

“whether higher BPL portends more TODs in patients with similar mean SBP” – unclear, please revise. Higher than what? Similar to what?

“DBPL groups did not defer in any of the cardiac parameters” – do the authors mean “differ”?

“Tatasciore et al.[19] assessed BPL as part of their study”. This can be shortened and merged to one sentence.

 “mean age 51.3±10.8 years, 49.5% men” – decimals for age and percentages can be omitted (the authors only include decimals half the time). Please revise throughout.

“…participants in the third tertile were slightly younger (mean age 50.0 vs. 52.2) but had multiple more cardiovascular risk factors including higher BMI, male sex, smoking status and…” – this seems irrelevant for the present review particularly if this finding of younger age is only seen in this study and is likely a chance finding.

The thresholds for BLP for each study and awake/sleep hours can be omitted from the main text as it is reported in Table 1.

Table 1

”models including 24-h BMP” – should probably be “models including 24-h MBP”.

Cardiovascular Outcomes

“An SBPL lower than this value had a negative predictive value of 98% in this cohort.” – was multiple regression used in the study, was the effect of SBPL adjusted for mean SBP or other BP measure?

“in a significant and meaningful way” – could be shortened to just “significantly”? The following sentence seems redundant.

“65% Flemish” – this and other minor details of the individual studies can be omitted.

“The risk of incident AF increased significantly in higher quartiles” – please revise, e.g. at the ?two? upper quartiles.

“CCD with the 24-hours SBPL (r≈0.45)” – was this association similar to the association with mean/systolic BP?

“aged 60-75 and” – “years” is missing, please revise throughout.

“The results for these measures were on a par with those for LVH.” – please rephrase and just state the results. Similar for the next study.

Discussion

“widespread use” – again, BPL is not in widespread use, there is no clear evidence, and is not recommend by guidelines.

“and has become a standard part of the ABPM report.” – as above, this is irrelevant.

“obviously the older technique of fixed awake and sleep time windows could lead to a mislabeling of readings, affecting the results” – redundant, can be shortened to e.g. “may have affected the results”.

“to a very relevant and sizeable group of patients” – seems redundant.

“but hopefully this challenge would be undertaken” – redundant.

“it can be summarized that significant and meaningful associations exist between BPL and the studied outcomes” – no, it seems to me that BPL was only slightly better than MBP in a few smaller studies whereas most larger studies found no effect of the addition of BPL.

The studies seem to make many comparisons for the different variables of BPL, SBPL, DBPL, night time, day time etc. which could make for type 1 errors. This may be noted in the limitations.

“Disappointingly,” – redundant.

“While a few of the smaller studies report a specific BPL parameter to be an independent variable in multiple regression analyses, this was generally not reproduced in other papers.” – this seems to be the most important sentence of the paper but is hidden away deep in the discussion.

This should also be the cornerstone of the conclusion.

The majority and larger studies do not find a utility of BPL and this should be reflected in the conclusion.

It seems unlikely that new studies will change the overall negative findings in the literature. Further, it’s not likely that new interventional drug trials will use BPL as guidance for antihypertensive treatment.

References

The DOI of published articles can probably be omitted.

I have no conflicts of interest to declare.

There are minor grammatical issues.I have commented on some of them in my report.

Reviewer 5 Report

Currently, I think it is a sufficient paper with a low grade of priority about hypotesis of publication.

I'd like to suggest some modifications by starting on MAJOR POINTS of WEAKNESSES.

First of all, ENGLISH LANGUAGE must be greatly checked and amaliorated. Sentences such as "We present the first review on the associations of BPL with target organ damage (TOD) and clinical outcomes..." OR "The current literature does not provide sound support for the use of BPL in clinical decisions" should be totally re-considered and written.

Moreover, I think quality of table on pages 5-6 is poor. PLEASE VERIFY and MODIFY with consistent graphic improvements.

Finally, I think DISCUSSION might be implemented by reading some following articles and perhaps similar contents could be insert in references:

Am J Med. 2022 Sep;135(9):1043-1050. doi: 10.1016/j.amjmed.2022.05.007.

Hypertens Res. 2021 Nov;44(11):1548-1550. doi: 10.1038/s41440-021-00729-8.

Hypertens Res. 2018 Aug;41(8):553-569. doi: 10.1038/s41440-018-0053-1. 

Blood Press. 2012 Apr;21(2):97-103. doi: 10.3109/08037051.2012.641266.

With my best regards.

ENGLISH LANGUAGE must be greatly checked and amaliorated. Sentences such as "We present the first review on the associations of BPL with target organ damage (TOD) and clinical outcomes..." OR "The current literature does not provide sound support for the use of BPL in clinical decisions" should be totally re-considered and written.

Round 2

Reviewer 1 Report

I still support my previous opinions

Author Response

We have nothing to add to our originial response.

Reviewer 2 Report

No more suggestions

Author Response

We thank the reviewer.

Reviewer 3 Report

Dear Editor,

I carefully read the revised version of the manuscript that is significantly improved compared to the original version. I recommend its publication in the Journal.

Author Response

We thank the reviewer.

Reviewer 4 Report

I commend the authors for their revision, the manuscript is improved.

Title

According to my previous comment, the authors stated in their response that they have added “A literature review of current evidence” in the title but this is still missing in the manuscript.

Results

”LVMI correlated with ASPB, ADPB” – should be “ASBP, ADBP”. There are other instances of incorrect abbreviations in table 1. “awake ASPB and SBPL. ”

“multivariable model which included BPM was negligible” – is BPM a typo?

Conclusion

the available evidence fails to show it offers” – something missing “the available evidence fails to show that it offers” or “the available evidence does not indicate that it offers“.

There are only minor language issues.

Author Response

The corrections were made. We thank the reviewer again for the attention.

Reviewer 5 Report

Re-check post modifications.

I think, nowhere, it is suitable for considering publication by editors with a low/medium grade of priority.
Overall level of papaer is average.

Author Response

There is no suggestion for us to respond to.

Round 3

Reviewer 1 Report

No new comments

Author Response

Nothing to respond to.